# Acute Toxicity and Sublethal Effects of Lemongrass Essential Oil and Their Components against the Granary Weevil, *Sitophilus granarius*

**DOI:** 10.3390/insects11060379

**Published:** 2020-06-18

**Authors:** Angelica Plata-Rueda, Gabriela Da Silva Rolim, Carlos Frederico Wilcken, José Cola Zanuncio, José Eduardo Serrão, Luis Carlos Martínez

**Affiliations:** 1Department of Entomology, Federal University of Viçosa, Viçosa, MG 36570-000, Brazil; angelicaplata@yahoo.com.mx (A.P.-R.); zanuncio@ufv.br (J.C.Z.); 2Department of Crop Science, Federal University of Viçosa, Viçosa, MG 36570-000, Brazil; gabrielasrolim@gmail.com; 3Department of Plant Protection, Paulista State University, Botucatu, SP 18603-970, Brazil; cwilcken@fca.unesp.br; 4Department of General Biology, Federal University of Viçosa, Viçosa, MG 36570-000, Brazil; jeserrao@ufv.br

**Keywords:** gas chromatography, lethal effect, natural products, repellency, respiration rate, stored product pest

## Abstract

In the present work, we evaluate the toxic and repellent properties of lemongrass (*Cymbopogon citratus* (DC. ex Nees) Stapf.) essential oil and its components against *Sitophilus granarius* Linnaeus as an alternative to insecticide use. The lethal dose (LD_50_ and LD_90_), survivorship, respiration rate, and repellency on adults of *S. granarius* exposed to different doses of lemongrass oil and some of its components were evaluated. The chemical composition of the essential oil was found to have the major components of neral (24.6%), citral (18.7%), geranyl acetate (12.4%), geranial (12.3%), and limonene (7.55%). Lemongrass essential oil (LD_50_ = 4.03 µg·insect^–1^), citral (LD_50_ = 6.92 µg·insect^–1^), and geranyl acetate (LD_50_ = 3.93 µg·insect^–1^) were toxic to *S. granarius* adults. Survivorship was 99.9% in insects not exposed to lemongrass essential oil, decreasing to 57.6%, 43.1%, and 25.9% in insects exposed to LD_50_ of essential oil, citral, and geranyl acetate, respectively. The insects had low respiratory rates and locomotion after exposure to the essential oil, geranyl acetate, and citral. Our data show that lemongrass essential oils and their components have insecticidal and repellent activity against *S. granarius* and, therefore, have the potential for application in stored grain pest management schemes.

## 1. Introduction

Chemical synthetic insecticides are used to control insects in stored grain facilities. Phosphine is commonly used in noxious gas form to control stored product pests worldwide [1]. Other alternative chemical methods to fumigants consist of protectants with long residual efficacies that target a broad spectrum of species [2]. Insecticides such as pirimiphos-methyl, spinetoram, and spinosad are grain protectants and provide a rapid, lethal effect in stored product pests [3,4,5]. However, these insecticides cause environmental pollution [6], atmosphere ozone-depletion [7], toxic waste [8], have a long residual period of toxicity [9], and have documented insecticide resistance [1]. Among the alternative strategies to insecticides, the use of plant essential oils has been proposed for insect control in stored grains [10,11].

Plant essential oils have different properties such as biodegradability, selectivity to target pests, and can reduce the use of conventional insecticides [12,13]. Plant essential oils are volatile substances obtained from flowers, fruits, leaves, roots, and stems through steam or hydrodistillation. Plant essential oils are composited by alcohols, aldehydes, aromatic phenols, esters, ethers, ketones, oxides, and terpenoids (monoterpenes and sesquiterpenes), and determine the aroma of the donor plant [14,15]. They are used for the food industry [16], pharmacology [17], medicine [18], and agriculture [19]. Terpenoids have been documented to cause toxicity or repellency against some insects [12,20]. In addition, plant essential oils are an ecofriendly alternative to controlling stored product pests because they do not penetrate the insect cuticle and grains [21,22].

Essential oils and their components cause toxic effects in insects via contact, ingestion, or fumigation. In this context, indirect effects such as deterrence, feeding inhibition, and repellency have been studied [13,22,23]. These components act on the central nervous system, affecting acetylcholine, γ-aminobutyric acid, and octopaminergic receptors, as well as some respiratory pathways [24,25,26,27]. The efficacy of essential oils and their chemical components is described in coleopteran pests of stored grain, with successful results for *Acanthoscelides obtectus* Say (Coleoptera: Bruchidae) in response to exposure to mint oil [28], *Sitophilus granarius* Linnaeus (Coleoptera: Curculionidae) exposed to cinnamon oil [29], and *Tenebrio molitor* Linnaeus (Coleoptera: Tenebrionidae) exposed to garlic oil [21]. Within this chemical component group, essential oils are a broad-spectrum insecticide active on starches and storage pests [11].

The granary weevil, *Sitophilus granaries,* is a cosmopolitan insect pest of storage facilities and processing plants. *Sitophilus granarius* causes damage to beans, corn, sorghum, nuts, oats, peanuts, rice, and wheat [30,31]. This pest is controlled with synthetic insecticides, such as phosphine, which is highly efficacious against *S. granarius*. However, phosphine resistance has been reported in some populations of *S. granarius* [32].

Lemongrass, *Cymbopogon citratus* (DC. ex Nees) Stapf. (Poales: Poaceae), is an important source of chemical metabolites worldwide, with application to pest control. Toxic effects of lemongrass essential oil and terpenoid components have been demonstrated with success in agricultural pest control [33,34,35]. In *S. granarius*, the lethal and sublethal effects caused by different conventional insecticides have been demonstrated [36]; however, lemongrass essential oil and its main components might be used to improve integrated pest management (IPM) of *S. granarius*.

The effects of lemongrass essential oil and two major components on *S. granarius* mortality, survivorship, respiration rate, and behavioral repellent response were evaluated. This contributed to the understanding of how this bioinsecticide controls the granary weevil and how it has the potential to become an alternative to synthetic chemical insecticides.

## 2. Materials and Methods

### 2.1. Granary Weevils

A *S. granarius* population, resistant to phosphine, was obtained from the Department of Grain Sciences and Industry of the Kansas State University (Manhattan, KS, USA) and used to demonstrate susceptibility to lemongrass essential oil. The population was frequently checked for levels of phosphine resistance [37], and reared in the Institute of Applied Biotechnology for Agriculture (BIOAGRO) of the Federal University of Viçosa (UFV) in the county of Viçosa (20°45′ S 42°52′ W), State of Minas Gerais, Brazil. Larvae and adults of *S. granaries,* free of insecticide residues, were placed in glass bottles (1000 mL) maintained in an acclimatized room at 26 ± 1 °C, 65 ± 15% RH, and a 12:12 h (light:dark) photoperiod. These insects were fed on pasta and wheat grains ad libitum. Newly emerged *S. granarius* adults (from infested wheat grains) that were 24 hours old were used in the experiments.

### 2.2. Essential Oil

The essential oil of *Cymbopogon citratus*, isolated from fresh leaves and extracted by a hydrodistillation method (using a Clevenger-type apparatus) [38], was purchased from Bauru Distillery and Company (Catanduva, São Paulo State, Brazil).

### 2.3. Gas Chromatography(GC) Analysis

Quantitative analysis of the lemongrass essential oil was performed in triplicate on a Shimadzu gas chromatograph model GC-17A equipped with a flame ionization detector (FID; Shimadzu Corporation, Kyoto, Kansai, Japan), using chromatographic conditions: a fused silica capillary column (30 m × 0.22 mm) with a DB-5 bound phase (0.25 μm film thickness); column pressure 110 kPa; helium carrier gas at a flow rate of 1.8 mL min^−1^; injector temperature 205 °C; detector temperature of 260 °C; column temperature programmed to start at 40 °C (isothermal remaining for 2 min) and increased from 3 °C min^−1^ to 260 °C (isotherm remaining at 260 °C, for 10 min). A sample of 1 μL (1% *w/v* in dichloromethane) was injected, using split mode (split ratio 1:10).

### 2.4. Gas Chromatography/Mass Spectrometry (GC/MS) Analysis

The identification of lemongrass essential oil components was performed with GC/MS mass-coupled gas chromatograph (CGMS-QP 5050A; Shimadzu Corporation, Kyoto, Kansai, Japan). One µL of essential oil containing 1% dichloromethane was injected in the splitless mode (1:10 ratio). The gas carrier used was helium, with 1.8 mL^−1^ constant flow rate on an Rtx^®^-5MS fused silica capillary column (30 m, 0.25 × 0.25 mm; Restek Corporation, Bellefonte, PA, USA), using the Crossbond^®^ stationary phase (35% diphenyl and 65% dimethyl polysiloxane). The initial temperature of the injector and detector was 40 °C, for 3 min, with a temperature increase from 3 °C/min to 300 °C and held for 25 min. For mass spectrometer detection, an electron ionization mode with ionization energy of 70 eV was programmed to detect masses in the range of 29–600 Da. Lemongrass oil components were identified using their Kovats indexes from original literature [39,40,41], by comparisons of their mass spectra and retention times with those of (C_3_−C_24_) *n*-alkanes and mass spectral data deposited in the Wiley 07 Spectroteca and National Institute of Standards and Technology (NIST08 and NIST11) databases.

### 2.5. Toxicity Test of Lemongrass Essential Oil and Components

Terpenoids of lemongrass essential oil, including citral and geranyl acetate, were purchased from Merck KGaA (Darmstadt, Germany). Lemongrass essential oil, citral, and geranyl acetate were diluted in 10 mL of acetone to obtain six doses (1.56, 3.12, 6.25, 12.5, 25, and 50 µg·insect^−1^). Serial doses and a control (acetone) were used to determine the dose–response relationship and estimate lethal doses. Each dose solution (1 µL) was applied on the bodies of 50 newly-emerged (24-hour-old) *S. granarius* adults using a Hamilton microsyringe (model 7001, KH Hamilton Storage GmbH, Domat/Ems, Switzerland). The insects were placed individually in glass vials (20 × 100 mm), covered with a piece of organza, and fed on wheat grains. Three replicates of 50 weevils were used for each dose. The experimental design was completely randomized. The number of dead insects was recorded after 24 h of exposure. Insects were considered dead if unable to walk when prodded with a fine hair brush.

### 2.6. Time–Mortality Test

Adults of *S. granarius* were individualized in glass vials (20 × 100 mm) and exposed to the lethal doses (LD_50_ and LD_90_) of lemongrass essential oil and components determined in the dose–response relationship. Exposure procedures and conditions were the same as described in Section 2.5. The number of alive insects was recorded every 6 h for 2 d. Three completely randomized replicates were used with all essential oil and component doses. Acetone was used as a control.

### 2.7. Respiration Rate

Respiration rate of *S. granarius* adults was evaluated for 3 h after exposure to LD_50_ and LD_90_ essential oil and its components. The granary weevils treated with acetone were used as the control group. The respirometry measurement was detected with a TR3C CO_2_ analyzer (Sable System International, Las Vegas, NV, USA) and recorded by a data acquisition system (ExpeData, Sable System International) using the methodology adapted from previous studies [42,43]. For CO_2_ quantification, a *S. granarius* adult was placed in a respirometry chamber (25 mL) and the chamber was connected to a closed air system. Then, the gas in the respiratory chamber was pumped to the O_2_ and CO_2_ analyzers, and compared with those from the control. To quantify the CO_2_ produced inside each chamber, an airstream scrubbed compressed O_2_ via drietite/acarite column was pumped through the chamber at a flow of 100 mL min^−1^ for 2 min. *Sitophilus granarius* adults were weighed on a Shimadzu analytical balance model AY220 (Shimadzu Corporation, Kyoto, Japan) before and after the test. Sixteen completely randomized replicates were used to evaluate essential oil, components, and control.

### 2.8. Behavioral Repellency Response

Adults of *S. granarius* were placed in Petri dishes (90 mm diameter), with filter paper discs (Whatman^TM^, Fisher Scientific, Leicestershire, LE, UK) at the bottom of the plate used as arenas. Half of the arena was treated with 250 µL of lemongrass essential oil and their components at the LD_50_ or LD_90_, and the other half with acetone and air-dried for five min [44]. An *S. granarius* was released in the center of the arena and monitored for 10 min. Twenty (Males/females, 1 ratio) insects were used and the experimental design was completely randomized. Behavioral repellency was recorded using a Canon digital camcorder model XL1 3CCD NTSC (Canon, Tokyo, Japan) with a 16X video lens (ZoomXL 5.5–88 mm, Canon, Tokyo, Japan). The measurement of the distance walked and time spent on each half-arena were obtained with the aid of a video tracking system (ViewPoint Life Sciences, Montreal, Canada). Weevils that spent <1 min or 50% of the time in the half-arena treated with components were considered repelled or irritated, respectively [44,45].

### 2.9. Statistical Analysis

The toxicity data were submitted to Probit analysis to obtain a dose-mortality curve [46]. The time–mortality data were analyzed for survival analysis (Kaplan–Meier estimators, log-rank test) with the Origin Pro 9.1 software (OriginLab Corporation, Northampton, MA, USA). Respiration rate data were submitted to two-way ANOVA and Tukey’s HSD test (*p* < 0.05). Behavioral repellency response (walked distance and resting time) data were submitted to one-way ANOVA, and a Tukey’s HSD (*p* < 0.05) test was also used for comparison of means. Respiration rate and behavioral repellency response were arcsine-transformed to meet assumptions of normality and homoscedasticity. Statistical procedures were analyzed by SAS 9.0 software (SAS Institute, Campus Drive Cary, NC, USA).

## 3. Results

### 3.1. Essential Oil Components

Thirteen components were found in the lemongrass essential oil, which was 96.83% of the total composition (Figure 1, Table 1). These components were neral (24.6%), citral (18.7%), geranyl acetate (12.4%), geranial (12.3%), limonene (7.55%), camphene (4.70%), citronellal (3.21%), nonan-4-ol (3.19%), β-caryophyllene (2.58%), citronellol (2.24%), 6-metil-hept-5-en-2-one (1.79%), caryophyllene oxide (1.89%), and γ-muurolene (1.70%). The structures of the main terpenoid components found in the examined lemongrass essential oil are presented in Figure 2.

### 3.2. Toxicity Test

The dose–response model provided a good fit to the data (*p* > 0.05), allowing the determination of toxicological endpoints and confirming the toxicity of lemongrass essential oil and its components to *S. granarius* (Table 2). The LD_50_ of the essential oil was 4.03 µg·insect^−1^ (3.29–4.94 µg·insect^−1^). The bioassay indicated that geranyl acetate was the most toxic component, with an LD_50_ of 3.93 µg·insect^−1^ (3.25–4.77) µg·insect^−1^, followed by citral (LD_50_ = 6.92 µg·insect^−1^; range of 5.63–8.58 µg·insect^−1^). Both components were used in subsequent tests. Mortality was less than 1% in the control.

### 3.3. Time–Mortality Test

The survival of *S. granarius* exposed to LD_50_ of the components varied significantly (log-rank test, χ^2^ = 39.88, df = 3, *p* < 0.001). Survivorship was 99.9% in the control, dropped to 57.6% with lemongrass essential oil, 43.1% with citral, and 25.9% with geranyl acetate (Figure 3A). Survivorship of *S. granarius* exposed to lethal dose LD_90_ also showed significant differences (log-rank test, χ^2^ = 105.91, df = 3, *p* < 0.001). Survivorship was 99.9% in the control, decreasing to 14.1% with lemongrass essential oil, 7.43% with citral, and 6.58% with geranyl acetate (Figure 3B).

### 3.4. Respiration Rate

The respiration rate of *S. granarius* was influenced by exposure to lemongrass essential oil and its components at LD_50_ and LD_90_ (Figure 4). For LD_50_, respiration rates differed after 3 h of exposure (F_3,59_ = 8.83; *p* < 0.001). The highest mean respiration rate was observed in control insects (1.84 μL of CO_2_ h^−1^), followed by insects exposed to lemongrass essential oil (1.52 μL of CO_2_ h^−1^), citral (1.39 μL of CO_2_ h^−1^), and geranyl acetate (1.32 μL of CO_2_ h^−1^). Similar results were obtained with treatments at LD_90_; respiration rates differed after 3 h of exposure (F_3,59_ = 7.47; *p* < 0.001), with mean rates of 1.73 μL CO_2_ h^−1^ in the control, 1.46 μL CO_2_ h^−1^ in insects exposed to essential oil, 1.16 μL CO_2_ h^−1^ in insects exposed to citral, and 1.12 μL CO_2_ h^−1^ in insects exposed to geranyl acetate.

### 3.5. Behavioral Repellency Response

Representative walking tracks of *S. granarius* adults released into half-treated arenas are shown in Figure 5. The distances walked were longer in the control and LD_50_ insects than in the LD_90_-treated ones. The distances walked by *S. granarius* were shorter in the half-arenas treated with lemongrass essential oil (F_2,23_ = 11.62, *p* < 0.001), geranyl acetate (F_2,23_ = 9.59, *p* < 0.020), and citral (F_2,23_ = 8.24, *p* < 0.018) in comparison with control arena (Figure 6). The resting time was longer in the control than in the insects exposed to LD_50_ and LD_90_. Varied adult behavior was found in *S. granarius* exposed to lemongrass essential oil (F_2,23_ = 8.73, *p* < 0.001), geranyl acetate (F_2,23_ = 9.17, *p* < 0.001), and citral (F_2,23_ = 12.32, *p* < 0.001) (Figure 6).

## 4. Discussion

This study investigated the chemical composition of lemongrass essential oil and assessed the insecticidal and repellent activities of the essential oil and its terpenoids citral and geranyl acetate against *S. granarius* under laboratory conditions. The chemical quantitative and qualitative analyses revealed 13 components of lemongrass essential oil. The components present in larger quantities in the lemongrass are neral, citral, geranyl acetate, geranial, limonene, and camphene, which have been reported for this essential oil [39,40,41]. Terpenoids are secondary metabolites with several functions in plant physiology, cell membranes [47,48], and defense of plants against insects and pathogens, as demonstrated for more complex components [48,49]. In lemongrass essential oil, citral and geranyl acetate may have a neurotoxic effect on *S. granaries* with rapid lethality, as reported for other insects [49,50,51]. Although the mode of action of this essential oil and its components has not been fully elucidated, their toxic effects suggest a viable alternative for the management of stored product pests.

Insecticidal and repellent action of lemongrass essential oil and its terpenoids against *S. granarius* were found in bioassays under laboratory conditions. Lemongrass topically applied was toxic against *S. granarius* adults (LD_50_ = 4.03 µg·insect^−1^) and mortality increased in a dose-dependent manner, as also reported in other pests [52,53,54]. *Sitophilus granarius* adults exposed to high doses of lemongrass essential oil (LD_50_ and LD_90_) showed muscle contractions and changes in locomotion, and when exposed to LD_90_, paralysis without recovery. In this case, symptoms were consistent in *S. granarius*, confirming neurotoxicity. There is a set of results that point to effects on the nervous system of insect pests such as *Bemisia tabaci* (Gennadius) (Hemiptera: Aleyrodidae) [55], *Callosobruchus maculatus* Fabricius (Coleoptera: Chrysomelidae) [56], and *Trichoplusia ni* Hübner (Lepidoptera: Noctuidae) [50] after lemongrass essential oil exposure (by contact or fumigation). These data show that the topical application of different doses of lemongrass essential oil in small volumes is toxic against *S. granarius*.

Chemical components of lemongrass essential oil demonstrated toxic effects on *S. granarius* adults. Geranyl acetate has stronger contact toxicity (LD_50_ = 3.93 µg·insect^−1^) than citral (LD_50_ = 6.92 µg·insect^−1^). Higher doses of citral inhibited acetylcholinesterase in *Galleria mellonella* Linnaeus (Lepidoptera: Pyralidae) and octopamine in *Periplaneta americana* Linnaeus (Blattodea: Blattidae) [57,58]. Geranyl acetate has competitive inhibition of acetylcholinesterase in *Aedes aegypti* Linnaeus (Diptera: Culicidae) and other neurotoxic responses in *Musca domestica* Linnaeus (Diptera: Muscidae), leading to paralysis and death [59,60]. Our results show that, in the adult stage, *S. granarius* is susceptible to terpenoids from lemongrass components. Many plant essential oils have components that kill or repel insect pests [54,59], and in *S. granarius*, they can be an ecologically safe alternative to other toxic components.

In this study, a high variation in *S. granarius* survival is mediated by the interaction of the lemongrass essential oil, citral, and geranyl acetate with target sites in the nervous system [51]. Time periods to induce mortality in *S. granarius* by this essential oil and components were from 24 to 48 h. The low survivorship of this insect seems to be due to the rapid action of lemongrass essential oil, citral, and geranyl acetate, as observed in other coleopteran pests of grains such as *Oryzaephilus surinamensis* Linnaeus (Coleoptera: Silvanidae), *Rhyzopertha dominica* Fabricius (Coleoptera: Bostrichidae), and *Tribolium castaneum* Herbst (Coleoptera: Tenebrionidae) after plant terpenoid exposure [29,61,62]. In this study, the compared effects of the lemongrass essential oil and its components on *S. granarius* occurred at various time periods. The lower time-mortality of insects exposed to LD_50_ of lemongrass essential oil in comparison with its compounds citral and geranyl acetate may be due to the lower amount of the components in the essential oil blend, ranging, i.e., citral 18.5% and geranyl acetate 12.5%. The rapid insecticidal activity against *S. granarius* suggests that lemongrass essential oil and its components can be effective against this stored product pest. Thus, they may be a valuable alternative to synthetic chemical insecticides, especially in the management of pest populations that have developed resistance to chemical insecticides.

The lemongrass essential oil, geranyl acetate, and citral negatively affect the respiration rate of *S. granarius* up to 3 h after exposure, which indicates the physiological stress caused by the components. The respiration of insects is affected by the energy necessary for their metabolism to produce physiological defense against essential oils [27,43]. Different respiratory responses have been reported for other insects exposed to essential oils and components in *Podisus nigrispinus* Dallas (Heteroptera: Pentatomidae) [51], *Sitophilus zeamais* Motschulsky (Coleoptera: Curculionidae) [63], and *T. molitor* [21]. Low respiratory rates result in high physical conditioning damage because the energy is reallocated at the expense of physiological processes [42,43] with the potential to affect muscle activity, causing permanent paralysis [43,45]. Inhalation of fumigant essential oils is associated with insect respiration rate [45,51]. Our results show that *S. granarius* exposed to lemongrass essential oil, geranyl acetate, and citral had a decrease in the respiration rates, suggesting a possible fitness cost and energy reallocation.

The behavioral response tests show that lemongrass essential oil, geranyl acetate, and citral affect *S. granarius*. Some insect pests alter their locomotion when exposed to lemongrass essential oil and its components and avoided the toxic environments after the detection of the chemical components [21,22,42]. Plant essential oils have been claimed to disrupt the recognition of the substrate, which impairs the orientation and locomotor activity of insects [29,44,64]. According to the results here obtained, the odor of essential oil and its components is repulsive to *S. granarius*. Terpenoids cross the insect body barrier through the spiracles and trachea [22,44] and could lead to important consequences in the control of insect pests of stored grains [65]. Our results show that *S. granarius* adults are repelled by lemongrass essential oil, geranyl acetate, and citral, suggesting that the use of lemongrass essential oil and its components may introduce an innovative approach to control this pest through manipulation of its foraging and avoidance behavior.

## 5. Conclusions

This study shows the potential of lemongrass essential oil, citral, and geranyl acetate as an insecticide or repellent IPM approach to manage *S. granarius*. These compounds caused significant effects on the mortality, respiration depletion, and repellency in this pest of stored grains. Additionally, the insecticide effects of lemongrass essential oil can be due to the synergism of components and their ability to penetrate the insect body or through the respiratory system. Lemongrass essential oil, citral, and geranyl acetate have toxic and sublethal effects on *S. granarius* and can be an alternative to synthetic chemical insecticides.

## Figures and Tables

**Figure 1 insects-11-00379-f001:**
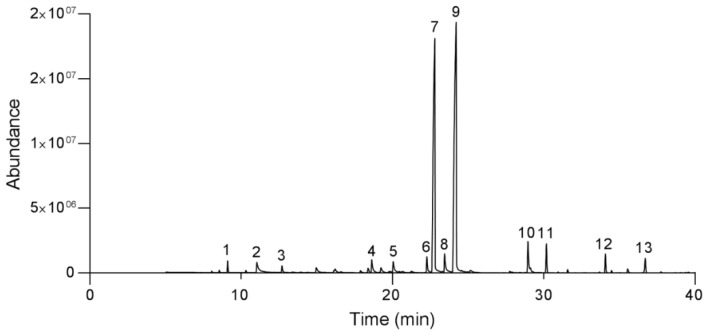
Gas chromatogram profiles of peak retention of components of the lemongrass essential oil: 6-methylhept-5-en-2-one (1), camphene (2), limonene (3), nonan-4-ol (4), citronellal (5), citronellol (6), neral (7), geranial (8), citral (9), geranyl acetate (10), β-caryophyllene (11), γ-muurolene (12), and caryophyllene oxide (13).

**Figure 2 insects-11-00379-f002:**
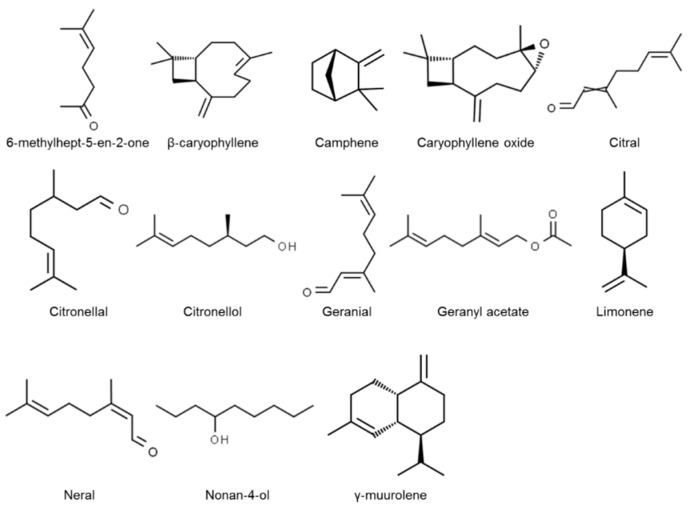
Chemical structure of main components identified in the lemongrass essential oil.

**Figure 3 insects-11-00379-f003:**
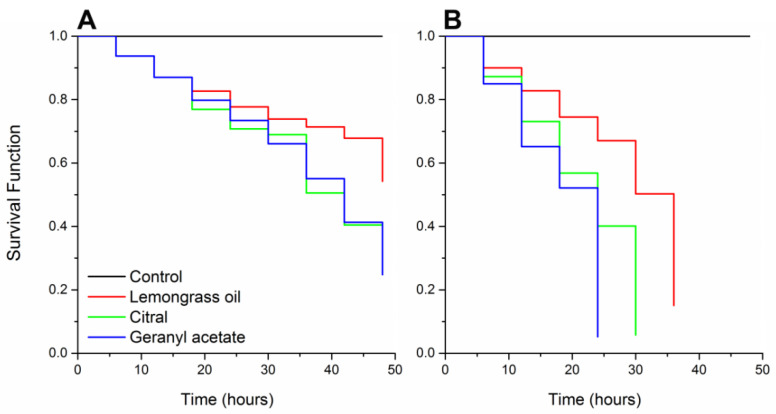
Survival curves of *Sitophilus granarius* adults exposed to lemongrass essential oil and its components, subjected to survival analyses using the Kaplan–Meier estimators’ log-rank test. Lethal dose (**A**) LD_50_ (χ^2^ = 39.88; *p* < 0.001) and (**B**) LD_90_ (χ^2^ = 105.9; *p* < 0.001).

**Figure 4 insects-11-00379-f004:**
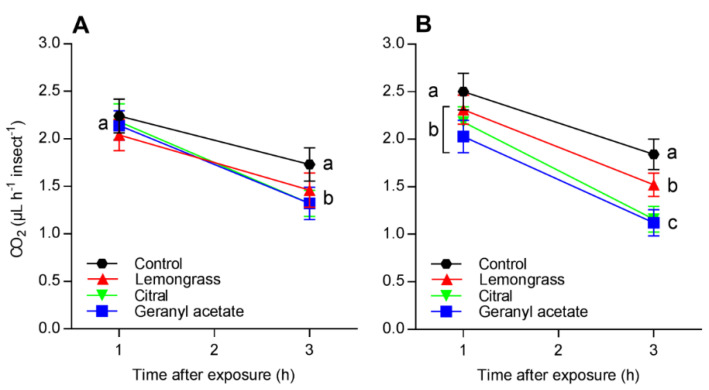
Respiration rate (mean ± SEM) of *Sitophilus granarius* adults after exposure to lemongrass essential oil, citral, and geranyl acetate, for levels (**A**) LD_50_ and (**B**) LD_90_. Treatments (mean ± SEM) differ at *p* < 0.05 (Tukey’s mean separation test).

**Figure 5 insects-11-00379-f005:**
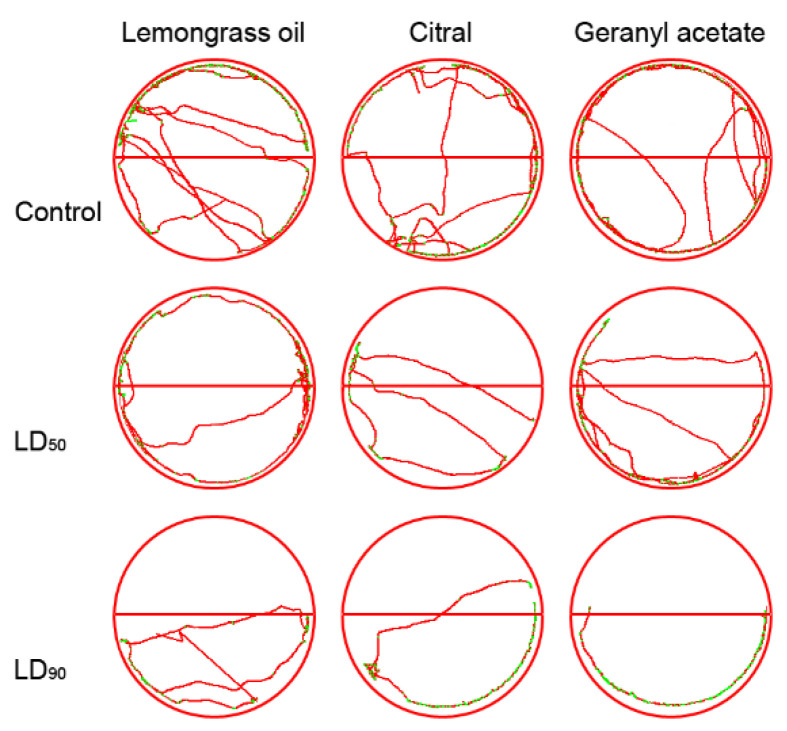
Representative tracks showing the walking activity of *S. granarius* over a 10-min period on filter paper arenas half-impregnated with lemongrass oil, citral, and geranyl acetate (upper half of each arena). Red tracks indicate high walking velocity; green tracks indicate low (initial) velocity.

**Figure 6 insects-11-00379-f006:**
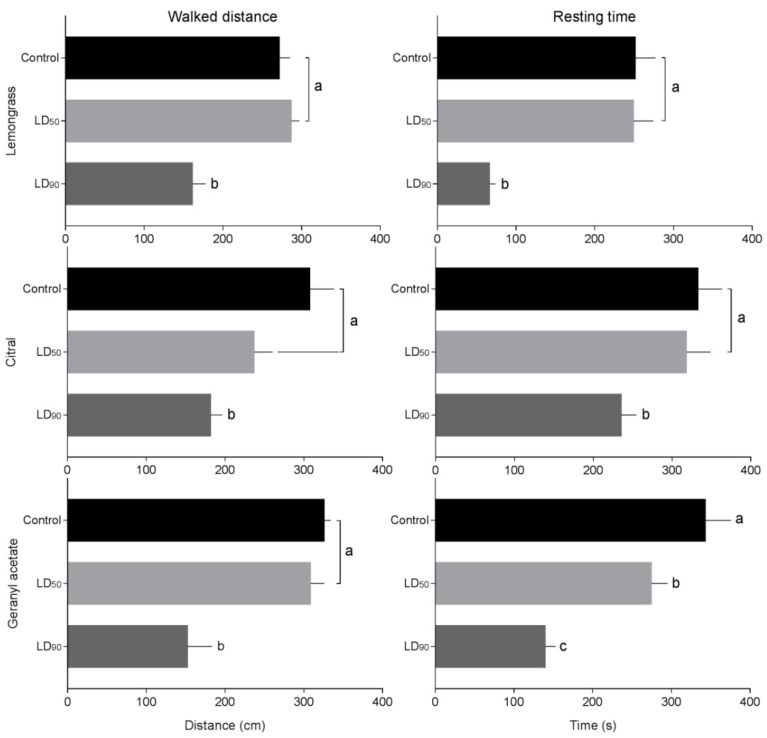
Distance walked and resting time (mean ± SEM) of *S. granarius* subjected to lemongrass essential oil, citral, and geranyl acetate (control, LD_50_, and LD_90_ estimated values) for 10 min. Treatments (mean ± SEM) differ at *p* < 0.05 (Tukey’s mean separation test). Values in the same column with different letters show significant differences by Tukey’s HSD test.

**Table 1 insects-11-00379-t001:** Chemical composition of lemongrass essential oil. Ri, retention indices; Rt, retention time; MC, mean composition (% area); MM, molecular mass; m/z, mass/charge ratio; ID, identification methods; KI, Kovats retention index on a DB-5 column and compared from the literature [39,40,41]; MS, mass spectra.

Peaks	Component	Ri	Rt	MC	MM	m/z	ID
1	6-methylhept-5-en-2-one	938	8.91	1.796	128	121.1	KI [39,40], MS
2	Camphene	958	10.8	4.709	130	108.1	KI [39,41], MS
3	Limonene	1030	12.4	7.552	136	94.1	KI [39,40], MS
4	Nonan-4-ol	1052	14.7	3.194	142	86.1	KI [39,41], MS
5	Citronellal	1125	18.5	3.213	154	121.1	KI [39,40,41], MS
6	Citronellol	1136	19.8	2.245	156	109.1	KI [39,40,41], MS
7	Neral	1174	22.1	24.65	156	95.1	KI [39,40,41], MS
8	Geranial	1179	22.5	12.36	152	109.1	KI [39,40,41], MS
9	Citral	1228	23.2	18.71	154	123.1	KI [39,40,41], MS
10	Geranyl acetate	1274	23.8	12.49	196	137.1	KI [39,40], MS
11	β-caryophyllene	1352	28.8	2.586	204	136.1	KI [39,40], MS
12	γ-muurolene	1435	29.9	1.706	204	133.1	KI [39,40], MS
13	Caryophyllene oxide	1494	33.8	1.893	220	204.1	KI [39,40], MS

**Table 2 insects-11-00379-t002:** Lethal doses of lemongrass essential oil and their components against *Sitophilus granarius* after 24 h of exposure, obtained from probit analysis (df = 5). The chi-square value refers to the goodness of fit test at *p* > 0.05.

Chemical Compound	No. Insects	Lethal Dose	Estimated Dose (µg·insect^–1^)	95% Confidence Interval (µg·insect^–1^)	Slope ± SE	χ^2^ (*p*-Value)
Lemongrass essential oil	150	LD_25_	2.388	1.799–2.955	2.956 ± 0.37	3.06 (0.63)
	150	LD_50_	4.039	3.293–4.943		
	150	LD_75_	6.830	5.535–9.006		
	150	LD_90_	10.95	8.408–16.23		
Citral	150	LD_25_	3.969	3.031–3.912	2.793 ± 0.33	6.77 (0.48)
	150	LD_50_	6.921	5.632–8.584		
	150	LD_75_	12.06	9.629–16.30		
	150	LD_90_	19.90	14.94–30.30		
Geranyl acetate	150	LD_25_	2.459	1.894–2.997	3.295 ± 0.42	2.47 (0.64)
	150	LD_50_	3.939	3.252–4.775		
	150	LD_75_	6.311	5.175–8.208		
	150	LD_90_	9.646	7.528–13.95		

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
