# Peer review of "Acute Toxicity and Sublethal Effects of Lemongrass Essential Oil and Their Components against the Granary Weevil, Sitophilus granarius"

_insects, 2020, doi:10.3390/insects11060379_

Round 1
Reviewer 1 Report
Article is interesting and the study was well conducted, but must be corrected before possible publication. Methodological errors and text editing are required:
1) Line 21 and 24: Correct “Chemical” and “S. granaries”.
2) Insert geographic coordinates in the section 2.1.
3) Detail how the essential oil was isolated.
4) Describe the oil yield and how the volatile compounds were quantified.
5) Draw the chemical structures of the main compounds of the essential oil and insert chromatograms.
6) Line 188: Insert RI from literature used for identifying the compounds. Discuss mass/charge ratio in the text.
7) Line 235 and 239. Figure 3 is duplicated. Review all citations of figures and tables.
8) Use S. granaries instead of the full scientific name after first description. The same for C. citratus.
Author Response
Response to Reviewer 1 Comments
Dear Dr. Nance Zhuo
Associate editor
Insects Journal
Please find enclosed our revised manuscript entitled “Acute Toxicity and Sublethal Effects of Lemongrass Essential Oil and Their Components against the Granary Weevil, Sitophilus granarius” (No. 798105_R1). The comments and suggestion provided were greatly appreciated and carefully considered for preparing the current version. The main changes are red-marked in the manuscript, and the point-by-point explanations to the comments provided follow below:
Reviewer #1
Article is interesting and the study was well conducted, but must be corrected before possible publication. Methodological errors and text editing are required:
1) Line 21 and 24: Correct “Chemical” and “S. granaries”.
2) Insert geographic coordinates in the section 2.1.
3) Detail how the essential oil was isolated.
4) Describe the oil yield and how the volatile compounds were quantified.
5) Draw the chemical structures of the main compounds of the essential oil and insert chromatograms.
6) Line 188: Insert RI from literature used for identifying the compounds. Discuss mass/charge ratio in the text.
7) Line 235 and 239. Figure 3 is duplicated. Review all citations of figures and tables.
8) Use S. granaries instead of the full scientific name after first description. The same for C. citratus.
I hope we addressed all the corrections indicated but we are ready to make any other that the reviewers consider necessary.
Sincerely,
Luis Carlos Martínez

Reviewer 2 Report
This paper evaluates the effects of lemongrass oil on the wheat weevil. The paper provided a strong background and methodology for doing the work. The results indicated strong effects of lemongrass oil on wheat weevils and the discussion appropriately addressed the results. The outcome of this research will be important in advancing lemongrass as an alternative IPM tactic form managing wheat weevils in stored grain settings. Overall, the paper was well written, but there are some grammatical/language issues that should be addressed.
Below are my comments, suggested edits, and recommendations.
Line 18: remove 'the', and I suggest adding that as a repellent it could be used complimentary to an insecticide.
Line 21: use a lowercase c for chemical.
Line 21: change 'has as' to 'was found to have the'.
Lines 22-26: these two sentences could be combined.
Line 24: italicize S. granarius.
Line 27: change 'exposition' to 'exposure'.
Line 28: change 'component' to 'components'.
Lines 34-35: Remove 'In this context,', start sentence with Phosphine.
Line 35: Remove 'insecticides'.
Lines 36-37: Suggest rewriting the sentence to say: Other alternative chemical methods to fumigants consist of protectants with long residual efficacy and target a broad spectrum of species.
Line 38: add 'such' between 'Insecticides as'.
Line 40: change 'residual long' to 'have a long residual period of toxicity'.
Lines 40-41: change 'insecticide resistance' to 'have documented insecticide resistance'.
Line 43: insert 'oils' between essential and have.
Lines 43-44: rewrite ...such as 'biodegradability, selectivity to target pests, low vertebrate toxicity, and can reduce the use of conventional insecticides."
Line 47: change 'determining' to 'and determine the'.
Line 49: change 'claimed to causes' to 'documented to cause'.
Line 51: change 'penetration into' to 'they do not penetrate into the'.
Line 53: change 'as' to 'such as'.
Line 57: change 'success results in' to 'successful results for'.
Line 64: change 'mainly phosphine that has high efficacy' to 'such as phosphine that is highly efficacous against this pest.'
Line 66: remove 'The' and start the sentence with Lemongrass.
Line 67: insert 'of' between source and chemical.
Line 67: change 'because' to 'with'.
Line 71: change 'control' to 'manage or improve IPM of'.
Line 73: change 'evaluated, contributing' to 'evaluated. This contributed'...
Line 74: change understand to understanding.
Line 74: change 'control of the' to 'controls the'.
Line 74: change 'weevil, as an alternative' to 'weevil and how it can serve to manage'.
Lines 78-79: Some may consider this to be pseudoreplication because you indicate here that the insects only came from one colony. I personally think that this is ok for the purposes of this paper, but future trials should consider multiple insect sources. A colony that has been established for a while may not be accurately representative of how wild populations may respond or behave.
Lines 78-79: It is important to include information about the source colony of insects, such as: how long they have been in this colony and if and how frequent wild additions are made to the colony.
Line 80. capitalize Larvae.
Line 92: insert 'vials were' before closed.
Line 93: insert 'insects were' before provided.
Line 93: delete 'of'.
Lines 94-95: you need to describe how you determined or confirmed that an insect was dead or considered morbid.
Line 102: change 'increase' to 'increased'.
Line 104: remove the ,.
Line 108: change 'A 1' to 'One'.
Line 109: lowercase for helium.
Line 117: remove the ,.
Line 124: insert 'a' between with and Hamilton.
Line 124: insert 'were' after insects.
Line 125: insert 'vials were' before closed.
Line 125: insert 'insects were' before provided.
Line 127: refer to previous comment about dead insects.
Lines 133-135: Rewrite as: Three completely randomized replicates were used with all essential oil and component doses.
Line 138: change 'their' to 'its'.
Line 143: change 'the gas into the' to 'gas in the'.
Lines 148-149: consider rewriting to: 'Sixteen completely randomized replicates were used to evaluate essential oil, components, and the control.'
Line 154: It could be important to understand the effects (or if there are any) based on sex. Here you do not mention any information about the number of males or females, and that is ok for this paper. I would consider looking at the effects on sex in future trials. Repellents can often be sex dependent in their activity.
Line 159: make Weevil plural and remove 'beetles'.
Line 159-160: I would strongly consider longer times in the arenas in future trials, or rather than a longer time in the arena, you could let the oil or components remain on the substrate for varying lengths of time before placing weevils on it. I would consider using grain as a substrate. The repellent effect could be lost quickly or may be different on a different substrate.
Line 162: For statistical analyses section, please provide your justification for all the transformations you used. Transformations are ok to use but you need to justify using them. Also, was anything specifically used to account for randomization, if so please state that.
Line 173: the survivorship of the control group that you mention here and elsewhere is incredible. It is so good that I can't believe it was that good without repeated attempts to maintain that level. Did you trash cohorts of insects when the control did not achieve a high level of survivability and repeat until you had the desired replication. If so, please mention that in the methods. If not, can you please describe more in the methods about the handling of the insects to achieve such minimal impact to their lives.
Line 175: rewrite ...oil, which was 96.83%...
Line 191: change 'were different' to 'varied significantly'.
Line 194: change 'showed differences' to 'also showed significant differences'.
Line 198: please state what the LD50 is of, mention the compound.
Lines 198-201: I am a little confused by the values you provide here. It looks like they match the 3h time but you mention both 1 and 3 hour. Please clarify this. At 1h the data do not look significant for either LD50 or LD90. Please clarify your wording.
Lines 206-208, Figure 1, Lines 192-196: You use % in lines 192-196 to describe the data but in the figure you use 'survival function'. Be consistent with what you call it.
Line 219: add 'conditions' after laboratory.
Lines 220-221: please clarify the wording 'according to dose'.
Lines 220-229: this section does not flow well. You mention other pests, then S. granarius, then back to other pests. Try to rewrite it to flow more logically and make a concise point.
Line 228: remove ','.
Line 250: change 'seems' to 'seem'.
Line 250: change 'alternative to control' to 'a viable alternative for the management of'.
Line 260: change 'repels' to 'repel' and 'pest' to 'pests'.
Line 263: remove 'the' before S. granarius.
Line 264: change 'system nervous' to 'the nervous system'.
Line 277: change 'response' to 'responses'.
Line 284: change 'has' to 'had a'.
Line 296: Rewrite: This study shows the potential of lemongrass essential oil, citral, and gernayl acetate as an insecticide or repellent IPM approach to manage S. granarius. These compounds caused significant effects on mortality...
Author Response
Response to Reviewer 2 Comments
Dear Dr. Nance Zhuo
Associate editor
Insects Journal
Please find enclosed our revised manuscript entitled “Acute Toxicity and Sublethal Effects of Lemongrass Essential Oil and Their Components against the Granary Weevil, Sitophilus granarius” (No. 798105_R1). The comments and suggestion provided were greatly appreciated and carefully considered for preparing the current version. The main changes are red-marked in the manuscript, and the point-by-point explanations to the comments provided follow below:
Reviewer #2
This paper evaluates the effects of lemongrass oil on the wheat weevil. The paper provided a strong background and methodology for doing the work. The results indicated strong effects of lemongrass oil on wheat weevils and the discussion appropriately addressed the results. The outcome of this research will be important in advancing lemongrass as an alternative IPM tactic form managing wheat weevils in stored grain settings. Overall, the paper was well written, but there are some grammatical/language issues that should be addressed.
Below are my comments, suggested edits, and recommendations.
Line 18: remove 'the', and I suggest adding that as a repellent it could be used complimentary to an insecticide.
Line 21: use a lowercase c for chemical.
Line 21: change 'has as' to 'was found to have the'.
Lines 22-26: these two sentences could be combined.
Line 24: italicize S. granarius.
Line 27: change 'exposition' to 'exposure'.
Line 28: change 'component' to 'components'.
Lines 34-35: Remove 'In this context,', start sentence with Phosphine.
Line 35: Remove 'insecticides'.
Lines 36-37: Suggest rewriting the sentence to say: Other alternative chemical methods to fumigants consist of protectants with long residual efficacy and target a broad spectrum of species.
Line 38: add 'such' between 'Insecticides as'.
Line 40: change 'residual long' to 'have a long residual period of toxicity'.
Lines 40-41: change 'insecticide resistance' to 'have documented insecticide resistance'.
Line 43: insert 'oils' between essential and have.
Lines 43-44: rewrite ...such as 'biodegradability, selectivity to target pests, low vertebrate toxicity, and can reduce the use of conventional insecticides."
Line 47: change 'determining' to 'and determine the'.
Line 49: change 'claimed to causes' to 'documented to cause'.
Line 51: change 'penetration into' to 'they do not penetrate into the'.
Line 53: change 'as' to 'such as'.
Line 57: change 'success results in' to 'successful results for'.
Line 64: change 'mainly phosphine that has high efficacy' to 'such as phosphine that is highly efficacous against this pest.'
Line 66: remove 'The' and start the sentence with Lemongrass.
Line 67: insert 'of' between source and chemical.
Line 67: change 'because' to 'with'.
Line 71: change 'control' to 'manage or improve IPM of'.
Line 73: change 'evaluated, contributing' to 'evaluated. This contributed'...
Line 74: change understand to understanding.
Line 74: change 'control of the' to 'controls the'.
Line 74: change 'weevil, as an alternative' to 'weevil and how it can serve to manage'.
Lines 78-79: Some may consider this to be pseudoreplication because you indicate here that the insects only came from one colony. I personally think that this is ok for the purposes of this paper, but future trials should consider multiple insect sources. A colony that has been established for a while may not be accurately representative of how wild populations may respond or behave.
Lines 78-79: It is important to include information about the source colony of insects, such as: how long they have been in this colony and if and how frequent wild additions are made to the colony.
Line 80. capitalize Larvae.
Line 92: insert 'vials were' before closed.
Line 93: insert 'insects were' before provided.
Line 93: delete 'of'.
Lines 94-95: you need to describe how you determined or confirmed that an insect was dead or considered morbid.
Line 102: change 'increase' to 'increased'.
Line 104: remove the ,.
Line 108: change 'A 1' to 'One'.
Line 109: lowercase for helium.
Line 117: remove the ,.
Line 124: insert 'a' between with and Hamilton.
Line 124: insert 'were' after insects.
Line 125: insert 'vials were' before closed.
Line 125: insert 'insects were' before provided.
Line 127: refer to previous comment about dead insects.
Lines 133-135: Rewrite as: Three completely randomized replicates were used with all essential oil and component doses.
Line 138: change 'their' to 'its'.
Line 143: change 'the gas into the' to 'gas in the'.
Lines 148-149: consider rewriting to: 'Sixteen completely randomized replicates were used to evaluate essential oil, components, and the control.'
Line 154: It could be important to understand the effects (or if there are any) based on sex. Here you do not mention any information about the number of males or females, and that is ok for this paper. I would consider looking at the effects on sex in future trials. Repellents can often be sex dependent in their activity.
Line 159: make Weevil plural and remove 'beetles'.
Line 159-160: I would strongly consider longer times in the arenas in future trials, or rather than a longer time in the arena, you could let the oil or components remain on the substrate for varying lengths of time before placing weevils on it. I would consider using grain as a substrate. The repellent effect could be lost quickly or may be different on a different substrate.
Line 162: For statistical analyses section, please provide your justification for all the transformations you used. Transformations are ok to use but you need to justify using them. Also, was anything specifically used to account for randomization, if so please state that.
Line 173: the survivorship of the control group that you mention here and elsewhere is incredible. It is so good that I can't believe it was that good without repeated attempts to maintain that level. Did you trash cohorts of insects when the control did not achieve a high level of survivability and repeat until you had the desired replication. If so, please mention that in the methods. If not, can you please describe more in the methods about the handling of the insects to achieve such minimal impact to their lives.
Line 175: rewrite ...oil, which was 96.83%...
Line 191: change 'were different' to 'varied significantly'.
Line 194: change 'showed differences' to 'also showed significant differences'.
Line 198: please state what the LD50 is of, mention the compound.
Lines 198-201: I am a little confused by the values you provide here. It looks like they match the 3h time but you mention both 1 and 3 hour. Please clarify this. At 1h the data do not look significant for either LD50 or LD90. Please clarify your wording.
Lines 206-208, Figure 1, Lines 192-196: You use % in lines 192-196 to describe the data but in the figure you use 'survival function'. Be consistent with what you call it.
Line 219: add 'conditions' after laboratory.
Lines 220-221: please clarify the wording 'according to dose'.
Lines 220-229: this section does not flow well. You mention other pests, then S. granarius, then back to other pests. Try to rewrite it to flow more logically and make a concise point.
Line 228: remove ','.
Line 250: change 'seems' to 'seem'.
Line 250: change 'alternative to control' to 'a viable alternative for the management of'.
Line 260: change 'repels' to 'repel' and 'pest' to 'pests'.
Line 263: remove 'the' before S. granarius.
Line 264: change 'system nervous' to 'the nervous system'.
Line 277: change 'response' to 'responses'.
Line 284: change 'has' to 'had a'.
Line 296: Rewrite: This study shows the potential of lemongrass essential oil, citral, and gernayl acetate as an insecticide or repellent IPM approach to manage S. granarius. These compounds caused significant effects on mortality...
I hope we addressed all the corrections indicated but we are ready to make any other that the reviewers consider necessary.
Sincerely,
Luis Carlos Martínez

Reviewer 3 Report
The article “Acute Toxicity and Sublethal Effects of Lemongrass Essential Oil and Their Components in the Wheat Weevil, Sitophilus granarius” is an interesting research paper describing toxic effect of lemongrass essential oil and two of its components against granary weevil. I highly recommend the authors to resubmit after improving/rewriting their paper. Overall, authors need to work on the English. Manuscript is not quite well written especially the Introduction section is difficult to follow. The study objectives and rational are not well written. Material and methods, discussion and conclusion sections need to be explained in more details. I highly recommend the authors to work with a professional writer to improve the English.
Line 3: Please change “in” to against
Line 4: I recommend the authors to change the wheat weevil to granary weevil throughout the paper
Line 21: lower case “c” in Chemical
Line 21: Please rewrite “essential oil has as major components”.
Line 22-24: Please rewrite “Lemongrass….adults.”. Stronger compare to what?
Line 35: Please rewrite. Methyl Bromide was a widely used fumigant, but it is phased out (in most countries) except for critical uses...
Line 27: change “exposition to” to “being exposed to”
Line 38: “such as” instead of “as”
Line 77: What is the history of these insects? Have they been exposed to any insecticide? Were they obtained from a laboratory-reared insecticide-susceptible colony?
Line 82: please explain how you obtained newly emerged adults?
Line 88: Did you mean “µg µL-1” ? and why did you use peanut and not their laboratory diet?
Lines 90-95: delete “Each dose….exposure.” It is written in lines 123-128.
Line 130: How many insects were used in each treatment and clearly mentioned what the treatments are.
Line 188: Table 2: What was the purpose of looking at the chemical composition of the lemongrass essential oil? And why did you choose citral and geranyl acetate?
Line 84 and 119: Combine “toxicity test” sections together.
Line 150: explain in more details. Did you let the filter paper to dry and then release the insects? Please mention if adopted from previous study and add citation.
Line 223: I am confused “Sitophilus granarius adults exposed to high doses of lemongrass essential oil (LD50 and LD90) show muscle contractions, changes in the locomotion and when exposed to LD90 paralysis without recovery” How did you define dead insects? Did you see muscle contraction?
Line 235: avoid red and green color combination as color-blind people has trouble seeing it.
Line 167 and 236: Please be specific and name the response data rather than saying “behavioral repellency response data”. Did you analyze high velocity vs low velocity walking?
Line 239: It should be Figure 4.
Line 261-262: Add citation. You did not test for fumigant toxicity.
Line 263-264: Add citation.
Line292-294: Please rewrite.
Please expand the conclusion section. Explain in more details how your findings are important for controlling granary weevils since they are internal feeders and you just looked at the topical toxicity.
Author Response
Response to Reviewer 3 Comments
Dear Dr. Nance Zhuo
Associate editor
Insects Journal
Please find enclosed our revised manuscript entitled “Acute Toxicity and Sublethal Effects of Lemongrass Essential Oil and Their Components against the Granary Weevil, Sitophilus granarius” (No. 798105_R1). The comments and suggestion provided were greatly appreciated and carefully considered for preparing the current version. The main changes are red-marked in the manuscript, and the point-by-point explanations to the comments provided follow below:
Reviewer #3
The article “Acute Toxicity and Sublethal Effects of Lemongrass Essential Oil and Their Components in the Wheat Weevil, Sitophilus granarius” is an interesting research paper describing toxic effect of lemongrass essential oil and two of its components against granary weevil. I highly recommend the authors to resubmit after improving/rewriting their paper. Overall, authors need to work on the English. Manuscript is not quite well written especially the Introduction section is difficult to follow. The study objectives and rational are not well written. Material and methods, discussion and conclusion sections need to be explained in more details. I highly recommend the authors to work with a professional writer to improve the English.
Line 3: Please change “in” to against
Line 4: I recommend the authors to change the wheat weevil to granary weevil throughout the paper
Line 21: lower case “c” in Chemical
Line 21: Please rewrite “essential oil has as major components”.
Line 22-24: Please rewrite “Lemongrass….adults.”. Stronger compare to what?
Line 27: change “exposition to” to “being exposed to”
Line 35: Please rewrite. Methyl Bromide was a widely used fumigant, but it is phased out (in most countries) except for critical uses...
Line 38: “such as” instead of “as”
Line 77: What is the history of these insects? Have they been exposed to any insecticide? Were they obtained from a laboratory-reared insecticide-susceptible colony?
Line 82: please explain how you obtained newly emerged adults?
Line 88: Did you mean “µg µL-1” ? and why did you use peanut and not their laboratory diet?
Lines 90-95: delete “Each dose….exposure.” It is written in lines 123-128.
Line 130: How many insects were used in each treatment and clearly mentioned what the treatments are.
Line 188: Table 2: What was the purpose of looking at the chemical composition of the lemongrass essential oil? And why did you choose citral and geranyl acetate?
Line 84 and 119: Combine “toxicity test” sections together.
Line 150: explain in more details. Did you let the filter paper to dry and then release the insects? Please mention if adopted from previous study and add citation.
Line 223: I am confused “Sitophilus granarius adults exposed to high doses of lemongrass essential oil (LD50 and LD90) show muscle contractions, changes in the locomotion and when exposed to LD90 paralysis without recovery” How did you define dead insects? Did you see muscle contraction?
Line 235: avoid red and green color combination as color-blind people has trouble seeing it.
Line 167 and 236: Please be specific and name the response data rather than saying “behavioral repellency response data”. Did you analyze high velocity vs low velocity walking?
Line 239: It should be Figure 4.
Line 261-262: Add citation. You did not test for fumigant toxicity.
Line 263-264: Add citation.
Line292-294: Please rewrite.
Please expand the conclusion section. Explain in more details how your findings are important for controlling granary weevils since they are internal feeders and you just looked at the topical toxicity.
I hope we addressed all the corrections indicated but we are ready to make any other that the reviewers consider necessary.
Sincerely,
Luis Carlos Martínez

Reviewer 4 Report
Plata-Rueda et al
Insects
The manuscript details the toxicity and sublethal effects of lemongrass oil and some of the components of the oil. The authors do a nice job of setting up the reason for testing lemongrass oil or other essential oils and provide good background for the question. Overall, the manuscript needs a good editorial eye since there are extra or missing words throughout the manuscript.
Specific comments:
Line 43: Missing a word. Plant essential oils?
Line 44: Missing a word. Also they reduce?
Line 49: Terpenoids have been claimed to… Phrasing is odd. Revise.
Line 134-135: As control was used acetone should be “Acetone was used as a control.”
Line 171-173: Please describe these results in some detail. Were any of the doses significantly different from each other or did they all overlap? What was the highest and lowest? Just a basic description is needed.
Lines 181-185: From how you describe it here, it makes it seem like you tested all components as listed in the previous section. Please make a note of what components you actually tested.
Line 192-193: It seems that there might be protective effects within the lemongrass oil that disappear with just citral or geranyl acetate. Can you describe this in the discussion?
Line 198: Showed higher respiration rates than what? This phrasing is confusing. Are you comparing the lemongrass, citral, and geranyl acetate to control?
Lines 210-216: I don’t know what resting time or varied adult behavior means. Please define in the methods and again in the results.
Figure 3. The labeling is somewhat misleading. Can you move the Lemongrass Oil, Citral, and Geranyl Acetate labels under the first row of circles so that they don’t include the control tracks?
Lines 245-248: Remind us what the terpenoids in your components list are.
Lines 250-251: Which seems to be alternative to control stored products… Odd phrasing, please revise.
Line 257: Should a consumer be worried about the toxic effects on things like the mouse?
Lines 258-259: I thought you conducted the research on contact to the essential oil and not inhalation. You have a partial components of this in the behavioral assay but not throughout the manuscript.
Line 261: How do you know this is through inhalation? How did you show this in the manuscript?
Author Response
Response to Reviewer 4 Comments
Dear Dr. Nance Zhuo
Associate editor
Insects Journal
Please find enclosed our revised manuscript entitled “Acute Toxicity and Sublethal Effects of Lemongrass Essential Oil and Their Components against the Granary Weevil, Sitophilus granarius” (No. 798105_R1). The comments and suggestion provided were greatly appreciated and carefully considered for preparing the current version. The main changes are red-marked in the manuscript, and the point-by-point explanations to the comments provided follow below:
Reviewer #4
The manuscript details the toxicity and sublethal effects of lemongrass oil and some of the components of the oil. The authors do a nice job of setting up the reason for testing lemongrass oil or other essential oils and provide good background for the question. Overall, the manuscript needs a good editorial eye since there are extra or missing words throughout the manuscript.
Specific comments:
Line 43: Missing a word. Plant essential oils?
Line 44: Missing a word. Also they reduce?
Line 49: Terpenoids have been claimed to… Phrasing is odd. Revise.
Line 134-135: As control was used acetone should be “Acetone was used as a control.”
Line 171-173: Please describe these results in some detail. Were any of the doses significantly different from each other or did they all overlap? What was the highest and lowest? Just a basic description is needed.
Lines 181-185: From how you describe it here, it makes it seem like you tested all components as listed in the previous section. Please make a note of what components you actually tested.
Line 192-193: It seems that there might be protective effects within the lemongrass oil that disappear with just citral or geranyl acetate. Can you describe this in the discussion?
Line 198: Showed higher respiration rates than what? This phrasing is confusing. Are you comparing the lemongrass, citral, and geranyl acetate to control?
Lines 210-216: I don’t know what resting time or varied adult behavior means. Please define in the methods and again in the results.
Figure 3. The labeling is somewhat misleading. Can you move the Lemongrass Oil, Citral, and Geranyl Acetate labels under the first row of circles so that they don’t include the control tracks?
Lines 245-248: Remind us what the terpenoids in your components list are.
Lines 250-251: Which seems to be alternative to control stored products… Odd phrasing, please revise.
Line 257: Should a consumer be worried about the toxic effects on things like the mouse?
Lines 258-259: I thought you conducted the research on contact to the essential oil and not inhalation. You have a partial components of this in the behavioral assay but not throughout the manuscript.
Line 261: How do you know this is through inhalation? How did you show this in the manuscript?
I hope we addressed all the corrections indicated but we are ready to make any other that the reviewers consider necessary.
Sincerely,
Luis Carlos Martínez

Round 2
Reviewer 1 Report
All suggestions were made.
Author Response
ok.

Reviewer 4 Report
The authors appear to have addressed my comments sufficiently. I have no further comments at this time.
Author Response
ok
